

# Impact of Asian aerosols on the summer monsoon strongly
# modulated by regional precipitation biases
Zhen Liu[1,2], Massimo A. Bollasina[2], Laura J. Wilcox[3]
[1]Earth, Ocean and Atmospheric Sciences (EOAS) Thrust, Function Hub, The Hong Kong University of Science and
Technology (Guangzhou), Guangzhou, China
[2]School of Geosciences, University of Edinburgh, UK
[3]National Centre for Atmospheric Science, Department of Meteorology, University of Reading, Reading, UK
Correspondence to Zhen Liu (henryzhenliu@hkust-gz.cn)



**Abstract.** Reliable attribution of Asian summer monsoon variations to aerosol forcing is critical to reducing uncertainties in future projections of regional water availability, which is of utmost importance for risk management and adaptation planning in this densely populated region. Yet, simulating the monsoon remains a challenge for climate models which suffer from long-standing biases, undermining their reliability in attributing anthropogenically-forced changes. We analyse a suite of climate model experiments to identify a link between model biases and monsoon responses to Asian aerosols, and the physical mechanism underpinning this link, including the role of large-scale circulation changes. The aerosol impact on monsoon precipitation and circulation is strongly influenced by a model's ability to simulate the spatial distribution and temporal variability of the climatological monsoon winds, clouds and precipitation across Asia, which critically modulates the magnitude and efficacy of aerosol-cloud-precipitation interactions, the predominant driver of the total aerosol response. There is a strong interplay between South and East Asia monsoon precipitation biases and their relative predominance in driving the overall monsoon response. We found a striking contrast between the early and late summer aerosol-driven changes ascribable to opposite signs and seasonal evolution of the biases in the two regions. A realistic simulation of the evolution of the large-scale atmospheric circulation is crucial to realise the full extent of the aerosol impact over Asia. These findings provide important implications to better understand and constrain the diversity and inconsistencies of model responses to aerosol changes over Asia in historical simulations and future projections.

## 1 Introduction

The Asian summer monsoon is one of the key components of the global atmospheric circulation, providing critical water resources to more than 60% of the world's population. Because of this reliance, even small changes in the spatio-temporal characteristics of the monsoon represent a significant hurdle for the local population. Yet, despite considerable efforts, simulating the monsoon remains a long-standing challenge for climate models as some biases have persisted for decades, such as the deficient rainfall over central India and excess wetting over eastern China (Sperber et al., 2013; Liu et al., 2021). The existence of these large and widespread biases not only decreases the confidence in the modelled monsoon and associated physical mechanisms (Yang et al., 2019; Jiang et al., 2020; Rajendran et al., 2022; Liu et al., 2022) but also represents a major cause of the large inter-model spread in historical monsoon evolution (Zhou et al., 2019; Guilbert et al., 2023). Moreover, these biases are likely to hinder reliable monsoon projections with critical implications for water management and planning across Asia and subsequent impacts on agriculture and economy (Zhou et al., 2019; Cao et al., 2020; Wang et al., 2020; Pillai et al., 2021). In particular, model biases introduce large uncertainties in our ability to separate externally-forced from internally-generated monsoon variability, preventing robust attribution of rainfall changes to specific drivers, including the extent to which recent and near-future trends are driven by anthropogenic aerosols (Wilcox et al., 2015; Dai et al., 2022).




Anthropogenic aerosols represent the largest uncertainty in quantifying the total anthropogenic forcing on climate since
the pre-industrial era (Andrews and Forster, 2020). Aerosols exert an overall cooling effect on climate by modulating
solar radiation via absorption and scattering, as well as by acting as cloud condensation and ice nuclei and thus altering
could albedo and lifetime and precipitation processes (Boucher et al., 2013). Asia has the largest present-day
anthropogenic aerosol burden as rapid urbanization and economic development have drastically increased aerosol
emissions and loading since the 1950s (Lin et al., 2016a). China has recently implemented strong pollution control policies,
which has substantially reduced aerosol emissions since 2013 (e.g., 59% for sulfur dioxide and 28% for black carbon
during 2013–2017; Zheng et al., 2018). Yet, Asia will still experience the highest aerosol loading in the world over the
coming decades as projected by the different future socioeconomic pathways used in the Coupled Model Inter-comparison
Project Phase 6 (Lund et al., 2019).

Aerosols have been found to play a key role in driving the observed decreasing trend in Indian summer rainfall (Chung
and Ramanathan, 2006; Lau and Kim, 2010; Bollasina et al., 2011; Guo et al., 2015) and the southern-flood-northern-
drought (SFND) pattern over East Asia (Menon et al., 2002a; Guo et al., 2013; Song et al., 2014a; Yu et al., 2016; Tian
et al., 2018) during the late 20[th] century. Studies that have separately investigated the impact of regional (Asian) and
remote (outside Asia) emissions have found the former to be fundamental to explaining the observed monsoon changes,
but with the latter also providing an important contribution (Cowan and Cai, 2011; Ganguly et al., 2012; Bollasina et al.,
2014; Dong et al., 2016). In particular, South and East Asian aerosols separately exert a strong influence on both the
South and East Asian monsoons, with contrasting, if not opposite, changes as well as strong non-linear interactions
between the responses to individual emission sources (Singh et al., 2019; Sherman et al., 2021; Herbert et al., 2022; Liu
et al., 2023).

At larger scale, the Asian monsoon march is linked to the evolution of semi-permanent features of the tropical and
extratropical atmospheric circulation, such as the western Pacific subtropical high (Zhang et al., 2005) and the Mascarene
high in the southern Indian Ocean (Vidya et al., 2020). The monsoon and the large-scale circulation are affected by
anthropogenic aerosol forcing, resulting in complex and intertwined interactions between externally and internally forced
variability (Deser et al., 2012; Huang et al., 2020a; Zha et al., 2022). Understanding the interplay between the Asian
monsoon and the large-scale circulation outside Asia and the extent to which concurrent changes in the large-scale
circulation modulate the monsoon response to regional aerosol changes is thus beneficial to achieve better monsoon
simulations and more robust projections (An et al., 2012; Liu et al., 2021).





One approach that has provided valuable insights into the mechanisms of aerosol-monsoon interactions is the
decomposition of the response into two complementary components: a fast response involving atmospheric and land
surface adjustments but fixed sea surface temperature (SST), acting on a short time scale (few years), and a slow response
scaling with SST changes (Samset et al., 2016; Li et al., 2020; Zhang et al., 2021). The fast and slow components over
and around Asia show similar features under global and regional (Asian only) aerosol forcing. In the case of sulfate
aerosols, the total response over Asia and downwind Pacific regions shows substantial precipitation decreases, with the
fast component featuring negative anomalies over land and positive ones over the adjoining ocean. While the monsoon is
a fully atmosphere-ocean coupled system, recent studies have found rapid adjustments to be of fundamental importance
in explaining inter-model differences in the aerosol response (Fläschner et al., 2016; Liu et al., 2018; Zanis et al., 2020).
Building on the above considerations, this study aims to identify a link between model biases and monsoon response to
Asian aerosols, and the underpinning physical mechanism, including the role of large-scale circulation changes outside
Asia and SST changes. The rest of the manuscript is organized as follows: Details of model experiments and analysis
methods are provided in Section 2. Section 3 examines the influence of precipitation biases on the climate response to
Asian aerosol perturbations and describes the underlying mechanism. Conclusions follow in Section 4.
**2 Data and methods**
The primary dataset analysed consists of simulations conducted with the Met Office Unified Model (MetUM) HadGEM3-
Global Atmosphere version 7.1 (GA7.1) at N96 horizontal resolution (1.875° × 1.25°) and with 85 vertical levels
extending up to 85 km (Walters et al., 2019). The GLOMAP modal aerosol scheme is used to represent aerosol processes,
including a representation of both aerosol-radiation and aerosol-cloud interactions (see Mann et al., (2010) and Bellouin
et al. (2013) for more details). GA7.1 was used as the atmospheric component of the climate model participating in
CMIP6. Compared to the previous model version (GA7.0), GA7.1 has a smaller global-mean anthropogenic aerosol
effective radiative forcing (Walters et al., 2019).
A set of four experiments (see Table 1) is performed with GA7.1 for the period December 1991 to December 2012 with
prescribed daily observed SST from the European Centre for Medium-Range Weather Forecasts (ECMWF) Interim Re-
Analysis (Dee et al., 2011). The reference experiment (CONT) is driven by monthly-varying historical emissions of
anthropogenic aerosols and precursors following CMIP6 (Hoesly et al., 2018). CONTfA is identical to CONT except for
having anthropogenic aerosol emissions of sulfur dioxide ($SO_2$), black carbon, and organic carbon and biomass burning
emissions fixed at the year 1991 over Asia (10°–45°N, 60°–125°E, the purple box in Fig. 1c). The difference between
CONT and CONTfA represents the fast response to changes in Asian anthropogenic aerosols.



To separate regional and remote circulation adjustments to aerosol forcing, dynamical nudging (also known as Newtonian
relaxation) is applied by constraining horizontal winds towards ERA-I (Kooperman et al., 2012; Liu et al., 2021). We
conducted another pair of experiments (NUDG and NUDGfA, respectively) identical to CONT and CONTfA except for
nudging horizontal winds to ERA-I outside Asia (the region outlined in Fig. 1c). The difference, NUDG minus NUDGfA,
represents the local response to Asian aerosols in the absence of concurrent changes in the large-scale atmospheric
circulation outside Asia. Nudging is only applied above the planetary boundary layer (model level 12, or approximately
850 hPa) so that low-level winds can adapt to surface conditions (e.g., different topography with respect to ERA-I; (Liu
et al., 2021)). Comparing the differences between the free-running experiments (i.e., CONT – CONTfA) and the nudged
runs (i.e., NUDG – NUDGfA) enable us to determine the extent to which simultaneous adjustments in the large-scale
atmospheric circulation outside the region modulate the Asian monsoon response to changes in regional anthropogenic
aerosols. To account for the role of internal variability, all experiments consist of three ensemble members initialized
from different atmospheric conditions. Only the last 10 years of each experiment are analysed (i.e., 2003 – 2012) when
Asian aerosol emissions are at the maximum (Fig. 1a). The results are however largely unchanged if a longer analysis
window is chosen (e.g., 15-year averages) as anomalies display similar large-scale features, albeit of slightly smaller
magnitude (not shown). The statistical significance of ensemble-mean differences relative to model internal variability is
estimated using a 35-year HadGEM3-GA7.1 experiment where all forcing factors are set at pre-industrial (1850) levels.
After splitting the output from this simulation into 26 overlapping 10-year segments, the probability distribution of the
unforced 10-year means for a 3-member ensemble is computed by randomly selecting three of these segments without
repetition, for a total of 2600 samples. In turn, the probability distribution of the 10-year ensemble-mean differences is
calculated by randomly selecting two of the 2600 samples a total of 10000 times (e.g., Efron and Tibshirani, 1993), and
the 90% confidence interval is estimated as the range in which 90% of the samples fall.

Data from the Precipitation Driver Response Model Intercomparison Project (PDRMIP; Samset et al., 2016) is also
utilized to corroborate our findings from a multi-model perspective. Two experiments are considered: the baseline
simulation forced by present-day (year 2000) levels in aerosols and greenhouse gases emissions/concentrations, and one
identical to the baseline run except for having a ten-fold increase in sulfate aerosol emissions/concentrations over Asia
(10°–50°N, 60°–140°E). The geographical distribution of the baseline sulfate burden in the PDRMIP ensemble (Myhre
et al., 2017) is very close to that in the CONT-CONTfA difference (Fig. 1b) over Asia, with the latter also showing an
approximately 10-fold increase in $SO_2$ emissions since the early 1990s (Fig. 1a), which ensures a sound comparison
between the different simulations. The PDRMIP experiments were run for 15 years with fixed present-day SSTs and for
100 years in coupled mode ((Liu et al., 2018). The response to Asian aerosols is identified as the difference between the
perturbed and baseline simulation averaged over the years 6–15 for both fixed SST and coupled simulations. In the case
of fixed SST, this choice is consistent with previous studies (Samset et al., 2016; Myhre et al., 2017) and accounts for the



adjustment time to the step change in emissions from the baseline simulation. We chose the same averaging period also
for the coupled experiments for consistency with the nature of the transient response to time-evolving emissions examined
in this study. In this sense, the coupled experiments allow us to ascertain whether the findings are sensitive to "fast"
oceanic-mediated responses (i.e., air-sea interactions), thus excluding the contribution brought about by slow (i.e., multi-
decadal) oceanic adjustments pertaining to a fully equilibrated atmosphere-ocean climate system.

We also analyse the transient historical simulations with the MetUM HadGEM3-GC2 coupled model described in (Wilcox
et al., 2019). These consist of four-member ensemble runs with all historical forcings, and a companion experiment in
which aerosols over Asia (5°–47.5°N, 67.5°–145°E) are fixed at their 1971–1980 mean levels. The difference between
the two ensemble means across two 10-year periods (i.e., 1999–2008 and 1971–1980) is interpreted as the total transient
response to Asian aerosol changes. We choose the later period 1999–2008 when aerosol emission differences maximize
(see Fig. 1 in Wilcox et al., 2019) and are at a comparable magnitude to our HadGEM3-GA7.1 simulations. These
experiments allow us to ascertain the consistency between uncoupled and coupled transient settings.

In light of the strong seasonality of the precipitation response to aerosol changes and the partial compensation between
the monsoon response in the early and late summer (Bollasina et al., 2013), we examine monthly precipitation and
circulation changes in addition to the June–September seasonal means. The simulated climatological precipitation and
circulation are evaluated against the arithmetic mean of the Climate Prediction Center Merged Analysis of Precipitation
(Xie and Arkin, 1997) and the Global Precipitation Climatology Project (GPCP) version 2 (Adler et al., 2003)
precipitation observations (Wang et al., 2014) and the ECMWF Reanalysis v5 (Hersbach et al., 2020) sea-level pressure
and 850-hPa winds for the period 1981–2010, respectively. These datasets are also used to provide a broader interpretation
of the aerosol-driven simulated changes in the context of recent observed trends.

## 3 Results

### 3.1 Monsoon response to Asian aerosols

The temporal evolution of the seasonal-mean differences in aerosol emissions and total aerosol optical depth (AOD)
between CONT and CONTfA averaged over Asia (the area enclosed by the purple box in Fig. 1c) is displayed in Fig. 1a.
The rapid rise of AOD after 2002 is mostly due to the increase in $SO_2$ emissions as the similarity between the respective
time-series indicates. BC and OC emissions exhibit a comparatively minor increasing trend, while biomass burning
emissions show negligible changes. The spatial distribution of changes in column-integrated sulfate burden closely
resembles that of emission changes and is characterized by large increases over eastern China and northern India (Fig.



1b). The pattern of the seasonal AOD change follows that of sulfate loading, further indicating the primary contribution
of SO$_2$ emissions to the total aerosol amounts over the region. Positive AOD anomalies also extend eastward from China
to the northwestern Pacific, reflecting atmospheric transport of aerosols by climatological southwesterly winds. Seasonal
mean aerosol changes across Asia are thus dominated by sulfate aerosols, consistently with longer-term trends since the
1950s (Lund et al., 2019), which hints at a predominant role of SO$_2$ emissions in driving the response discussed below.

Fig. 2a shows the aerosol-driven summer precipitation changes. A band of excess rainfall stretches from southeastern
China and the South China Sea (SCS) across northern Indochina and the northern the Bay of Bengal (BOB) to northern
India, associated with a negative sea-level pressure anomaly and anomalous cyclonic flow centered over the northern
BOB (Fig. 2b). The simultaneous northwestward shift and strengthening of the Mascarene High over the equatorial Indian
Ocean leads to an enhanced cross-equatorial southwesterly flow over the western tropical Indian Ocean and subsequent
north-equatorial eastward moisture transport and precipitation increase across the basin (Fig. 2b and 2c), indicating an
intensified monsoon circulation. The anomalous wind then turns anticlockwise, bringing abundant moisture across the
BOB to northern India, Indochina, and southern China (Fig. 2c). Concurrently, anomalous dry westerlies over central
India lead to precipitation decrease, resulting in an approximately southwest to northeast oriented wet/dry rainfall dipole.

Over China, the widespread wetting to the south, together with the drying to the north, form a meridional dipole. The
dipole is accompanied by a marked anomalous anticyclone centered over the western subtropical Pacific and extending
further inland, suggesting a strong dynamical link with the rainfall anomalies via modulation of the climatological western
Pacific subtropical high (WPSH; Fig. 2a and 2b). On the southwestern flank of the anticyclone, anomalous southeasterlies
blow from the sub-tropical western Pacific across SCS and bring moisture to southern China (Fig. 2c). Here the flow
converges with the southwesterly winds from the Indian Ocean mentioned above, resulting in the abundant precipitation
increase. Moist southerlies further extend over eastern China and result in a positive, albeit of weak magnitude, moisture
convergence anomaly. This contrasts with the local precipitation deficit which, given also the modest evaporation
anomaly (not shown), appears to be associated with moisture divergence due to transient eddies whose contribution to
the total moisture flux convergence is relevant for the region (e.g., Seager et al., 2010; Li et al., 2018).

Examining these changes in a broader context, the aerosol-driven rainfall pattern displays, in its large-scale features, a
remarkable similarity, but opposite sign, to observations (Fig. S1). In particular, model and observations feature key
rainfall action centers of comparable magnitude and in similar geographical locations while of opposite polarity. For
example, observed changes show drying from northern India across the northern BOB to southeastern China, with wetting
over central and western India, northern China, and the western subtropical Pacific (Fig. S1a), in stark contrast to the
simulated anomalies shown in Fig. 2a. These precipitation anomalies over East Asia are associated with an anomalous





cyclone over the western subtropical Pacific (Fig. S1b; compared to the anomalous anticyclone in Fig. 2), leading to
oceanic moisture advection over northern China and dry northeasterlies over southern and eastern China. Anomalous
anticyclonic anomalies are seen over the northern BOB in contrast to a low over the Arabian Sea, which leads to excess
rainfall over central and western India and a deficit to the northeast and the northern BOB (opposite to the simulated
dipole in Fig. 2). Interestingly, the consistency between observed and simulated (sign-reversed) precipitation and sea-
level pressure patterns is also evident in CONT (Fig. S1c and S1d), albeit with some dissimilarities over land, while it is
less obvious when aerosol emissions are not evolving (Fig. S1e and S1f), particularly around the Indian subcontinent and
eastern China, resulting in an overall mixed signal. While this suggests a possible important role of aerosols in driving
the model anomalies, the opposite polarity of the aerosol-induced patterns compared to observations is puzzling and
warrants further investigation into the underpinning cause and physical mechanism.

Inspection of monthly precipitation and low-level circulation changes reveals a stark contrast over the Indian subcontinent
and adjacent ocean between the early and late monsoon season: increased precipitation and anomalous cyclonic flow over
the BOB in June, consistent with the seasonal mean, and decreased precipitation and anomalous anticyclonic winds over
India in September (Figs. S2 and S3). Rainfall and circulation anomalies in July display similarities to those in June, while
August shows a mixed pattern, with more spatially confined and smaller magnitude anomalies. Over East Asia, the June-
July precipitation anomalies, closely resembling the seasonal-mean changes, feature a zonal-elongated meridional dipole,
with wetting stretching from Indochina and southern China to the South China Sea, and drying to the north across central
and eastern China. Interestingly, the dipole reverses sign in September, accompanied by a southward displacement (i.e.,
the dipole nodal line moves from around 30°N to about 20°N), with widespread drying over southern Indochina and most
of the western subtropical Pacific and wetting to the north over northern Indochina and most of China. Consistently with
the comparison for the seasonal means, the sub-seasonal aerosol-driven simulated response patterns bear a strong
similarity, with opposite signs, to those observed (not shown).

These contrasting changes in the simulated aerosol-induced responses between the early and late summer, despite
negligible monthly variations in magnitude and spatial distribution of aerosol emissions across Asia, especially for $SO_2$
(not shown), and the consistently reversed polarity of their key centres compared to observations, suggest that different
mechanisms may underpin the responses throughout the season. This also suggests a possible link between long-term
changes and the underlying mean seasonal cycle, and the possibility of discrepancies between simulated and observed
characteristics in the latter to be the cause of the differences in the former. From a more general perspective, this also
highlights the importance of investigating and interpreting seasonal monsoon changes accounting for the pronounced sub-
seasonal variability in the response – an aspect usually overlooked in aerosol-monsoon research but particularly relevant
for attribution studies.



### 3.2 A mechanism linking model climatology to response


The accuracy of the simulated regional climate change signal and its attribution to anthropogenic drivers have been
suggested to be strongly dependent on the model performance in reproducing the corresponding mean climatological
conditions, which represent the baseline state on top of which changes occur (Matsueda and Palmer, 2011; Christidis et
al., 2013). Indeed, a link between model bias and corresponding response has been shown to hold, for example, in the
case of summer precipitation over Asia (Wilcox et al., 2015), global SST patterns and overlying rainfall changes (He and
Soden, 2016), tropical rainfall (Chadwick, 2016) and circulation (Zhou and Xie, 2015) extratropical stationary eddies and
their influence on tropical convection (Chen et al., 2018) and Arctic Ocean temperature (Park and Lee, 2021). Given that
state-of-the-art climate models still suffer from large and persistent biases in simulating magnitude and distribution of the
monsoon precipitation and circulation across Asia (Wilcox et al., 2020; Rajendran et al., 2022; Tong et al., 2022), it is
certainly plausible for these biases to exert a sizeable control on the aerosol-induced monsoon changes. Climatological
biases in climate models could lead to unrealistic projections of anthropogenic climate change and add further
uncertainties, for example due to their possible non-stationarity (Krinner and Flanner, 2018). Examining the sub-seasonal
evolution of the model bias could therefore provide insights into the simulated aerosol-induced monsoon response
described above, a topic that has been insufficiently addressed and possibly underappreciated so far.

Fig. 3a and 3b show the model precipitation bias relative to the mean of CMAP and GPCP in June and September,
respectively. In June, there is a clear anomalous meridional dipole over the Indian sector with rainfall excess over the
equatorial Indian Ocean and deficit over India and surrounding oceanic areas, including the BOB (Fig. 3a). This dipole
pattern is similar to that of the seasonal-mean bias commonly presented in both uncoupled and coupled models (Song and
Zhou, 2014; He et al., 2022; Rajendran et al., 2022). Particularly, the magnitude of the June dry bias over the Indian
subcontinent ($-2.8$ mm day$^{-1}$) is about 60% of the observed climatological amount (4.7 mm day$^{-1}$). The model is
excessively wet to the east over northern Indochina and most of China, particularly to the south, with predominant dry
anomalies over the China Sea. Interestingly, this bias pattern over continental Asia, and particularly the contrasting dipole
between (dry) India and (wet) northern Indochina/China, bears a close resemblance, with opposite sign, to the June
aerosol-induced simulated precipitation distribution discussed above (wetting over India and drying over northern
Indochina/China; Fig. 3b). Is there a mechanistic link between bias and aerosol-driven response?

Aerosol-cloud interactions have been found to play a fundamental role in modulating the Asian summer monsoon
response to anthropogenic aerosols, both in uncoupled and coupled experiments (Guo et al., 2015; Li et al., 2018).
Hydrophilic aerosols (e.g., sulfate) activated at a given supersaturation level can serve as cloud condensation nuclei and
increase the cloud droplet number concentration (CDNC). At constant cloud liquid water content, the increases in CDNC





reduce the cloud effective radius and enhance the cloud albedo (Twomey, 1974), exerting a cooling effect at the surface.
Meanwhile, the smaller cloud droplets reduce the collision/coalescence probability of droplets and thus weaken the
precipitation efficiency (Albrecht, 1989). Cloud effective radius, the critical variable linking aerosol emission changes to
cloud and precipitation variations, is proportional to the liquid water content at a given CDNC (Menon et al., 2002b).
While column water content also changes in response to aerosol variations (Sato et al., 2018; Wang et al., 2022), and thus
cause and effect are tightly intertwined at short time scales, the above hints at the possibility of baseline conditions to
modulate aerosol-cloud interactions and the subsequent monsoon response to aerosol changes, especially in the presence
of large model discrepancies in simulating the climatological distribution of atmospheric moisture (John and Soden, 2007;
Bastin et al., 2019; Han et al., 2022). The marked and abrupt shift in the atmospheric state accompanying the monsoon
onset and subsequent establishment across Asia is also by nature substantially affected and pre-conditioned by the
presence of anomalous conditions in the preceding spring months. In view of this and to better identify possible precursor
conditions leading to the marked aerosol-induced response in June, we examine the model anomalies in late spring.

It is worth noting that while the band of excess climatological rainfall over southern and eastern China is present for most
of the year, the magnitude of the bias undergoes a rapid increase from April until the peak in June and then decays from
July to September (Fig. S4). Also, the wet bias over eastern China is particularly spatially extensive in the spring (up to
50% of the climatology), while a weak, dry anomaly appears in the summer over the lower reaches of the Yangtze River.
Importantly, the widespread wet anomaly over China in April–May is largely collocated with the largest aerosol emission
sources, particularly $SO_2$ (Fig. S5a). The excess climatological moisture available over China provides favourable
conditions for the aerosol impact via aerosol-cloud interactions in addition to changes in radiation. In fact, the CONT–
CONTfA difference shows reduced shortwave clear-sky radiation at the surface, a simultaneous increase in cloud droplet
number concentration, and a decrease of the cloud-top effective radius (Fig. S5b–d). An anomalous anticyclone situated
over southeastern China (Fig. S5h), consistent with the pattern of aerosol forcing, leads to a meridional dipole in the water
content and precipitation response, with large and widespread wet anomalies over Indochina and the SCS and drying over
eastern China (Fig. S5e and S5f).

Conversely, the model underestimates the observed rainfall over the already dry pre-monsoon Indian subcontinent, with
a substantial dry bias over eastern India and the BOB (Figs. S4a and S4b). In response to the aerosol increase, there is a
clear reduction in clear-sky shortwave radiation over India, albeit secondary to that over China due to the smaller emission
changes, and a minor increase (decrease) in cloud droplet number concentration (cloud top effective radius) (Fig. S5b–
d). This indicates overall weak aerosol-radiation and aerosol-cloud interactions, resulting in negative, although very weak,
precipitation anomalies and associated mixed lower-tropospheric circulation response (Fig. S5f and S5h). It is worth
noting that the upper-level divergent outflow from the rainfall maximum anomaly over Indochina converges over



northeastern India where it subsides and generates a near-surface return flow, forming a system of closed and interacting
cells (Fig. S5g and S5h).

With the arrival and establishment of the monsoon over Asia in June, the simulated climatological precipitation increases
considerably over northern Indochina and southern China but only marginally over India (Fig. S6), resulting in a
substantial zonal precipitation dipole in the model bias across Asia, with marked dry anomalies over and around India
and wet conditions over Southeast Asia and southern China (Fig. 3a). There is also a dry anomaly over eastern China,
resulting from weak southwesterlies and stagnation of the monsoon front to the south (Liu et al., 2021). As in the spring,
the patterns of the precipitation bias and associated water content anomalies are important to understand the corresponding
aerosol-driven response. The anticyclonic circulation anomaly over eastern China strengthens and widens compared to
the earlier months from both aerosol-radiation and aerosol-cloud interactions (Fig. 3c and S5h), manifested in the
considerable reduction in surface clear-sky shortwave radiation (Fig. S7c), overall increased cloud droplet number
concentration, and decreased cloud effective radius over central and eastern China (Fig. 4a and 4b). Note the latter displays
positive anomalies over southern China, where enhanced easterlies along the southern flank of the anticyclone bring
moist-laden air from the western Pacific towards South Asia (Fig. 3c), leading to increased water availability and the
large precipitation excess there. Over India, the substantial climatological atmospheric water content deficit, seeing in the
ensuing large dry bias (Fig. 3a), strongly limits local aerosols to exert a sizable impact by markedly weakening the
magnitude of regional aerosol-cloud interactions (e.g., modest changes in could top effective radius in Fig. 4b). Regional
anomalies in the aerosol response are thus interpreted as remotely-induced by the large-scale circulation adjustment to
aerosol changes over China. Local aerosols and ensuing circulation and precipitation response are therefore tightly
coupled over Asia and linked by positive feedbacks, whereby an initial aerosol-induced anomaly in precipitation
subsequently acts to reinforce the anomalous pattern by regional circulation adjustments. For example, the deep ascent
and upper tropospheric divergent outflow associated with the excess precipitation over the BOB and Indochina bifurcates
with the primary branch converging and subsiding over northeastern China (reinforcing the local anticyclone) and the
secondary branch over the southeastern Philippines Sea where dry anomalies are found and are part of the northwestward
rainfall shift (Fig. S8a–c).

As summer progresses, the simulated climatological precipitation reaches its peak over India, while it retreats markedly
over China (Fig. S7a and S7b). This, together with the anomalies set up as part of the aerosol response in the earlier part
of the season (Fig. 3b), leads to enhanced moisture availability over South Asia compared to the earlier months, while
also partially alleviating the reduced, but persisting, model dry bias there (Fig. 3e). Conversely, over China, the moisture
deficit from the aerosol-weakened monsoon circulation (Fig. 3b), as well as the rapid monsoon demise in the simulated
climatology (Fig. S7e), contribute to lessening the degree of interaction between aerosols and clouds and precipitation.



As a result, aerosol-cloud interactions over South Asia are more effective compared to the early summer, and the
continental-scale simulated response is predominantly driven by aerosol-induced anomalies over South Asia. Associated
with a decrease in cloud top effective radius (Fig. 4f), negative precipitation anomalies appear over South Asia from
August (Fig. S3c), with maximum peak in September (Fig. 3f). Correspondingly, the lower-tropospheric circulation
features an anomalous anticyclone, with westerly winds over northern India and the BOB (Fig. 3g). The flow turns
southwesterly over northern Indochina, bringing more moisture to eastern China (leading to increased could top effective
radius in Fig. 4f) and increasing precipitation which forms a zonal dipole with the rainfall decrease over the north-
equatorial western Pacific (Fig. 3f and 3h). The associated anomalous western Pacific anticyclone weakens and shifts
eastward (Fig. 3g). The large-scale anomalous circulation pattern is characterized by the mid-tropospheric vertical motion
and divergent outflow over southern China, and upper-level convergence and subsidence over South Asia and the north-
equatorial western Pacific (Fig. S8d–f), which further attests for the strong coupling across the region.
**3.3 Aerosol response in nudged simulations**
The Asian monsoon response to aerosol changes discussed above entails large-scale atmospheric circulation adjustments
extending beyond the Asian region. It is therefore interesting to understand whether and the extent to which they
contribute to driving the regional response. Constraining the large-scale circulation outside Asia to observations allows
us to isolate the effect of remote (i.e., outside Asia) circulation changes in generating the monsoon response to Asian
aerosols.

The AOD differences between the pair of nudged simulations (Fig. S9) resemble those shown in Fig. 1 despite the
considerably different circulation and precipitation anomalies (see Fig. 3c, 3g, 5c, 5g), indicating that the AOD
distribution is predominantly influenced by emissions changes rather than by aerosol transport and removal processes.
The spatial patterns of both the June and September precipitation biases in NUDG, where horizontal winds outside Asia
are nudged to ERA-I, are overall very similar to those in the control simulation (cf. Fig. 5a and 5e). Sub-regional
differences in the magnitude of the bias between the two sets of experiments are however noticeable (e.g., the dry bias is
markedly reduced over India, whereas southeastern China is wetter, compared to CONT), indicating that an improved
representation of the remote circulation can potentially reduce the precipitation bias in some areas but not necessarily
across the entire domain (Liu et al., 2021).

The June precipitation response to aerosol changes features an approximately meridional dipole, with widespread drying
from north-eastern India to southern China and wet anomalies over central India, the BOB, and parts of the South China





Sea (Fig. 5b). Compared to the free-running simulations, precipitation anomalies are of much smaller magnitude and
mostly confined within Asia and the neighbouring oceanic areas, without a significant aerosol signature downstream (e.g.,
over the equatorial Indian Ocean and the subtropical Pacific). This is expected as the atmospheric circulation above the
planetary boundary layer is nudged outside Asia, and attests to the key role of large-scale circulation adjustments in
realizing the aerosol impact. Consistently with the link between rainfall bias and response to aerosol forcing found in the
free-running simulations, wetter climatological conditions over China and a reduced dry bias over India translate into
more efficient aerosol-cloud interactions over both regions (Fig. 5a). As a result, the ensuing precipitation response, while
bearing similarities with that in Fig. 3 and thus on the driving role of Chinese aerosol emissions, also shows noticeable
differences: the drying over China is more spatially extensive, particularly to the south, while the wetting over the Indian
sector is mainly confined to the northern BOB (Fig. 5b). While sign and pattern of the aerosol-induced response are
consistent with the bias pattern, the generally weak anomalies are a result of the unchanged large-scale circulation outside
Asia in the nudged experiments. For example, Fig. 5c shows a pattern of sea level pressure anomalies which resembles
that shown in Fig. 3c, but with much smaller gradients and mostly confined to Asia only. In particular, there is only a
very weak westerly flow across the north-equatorial Indian Ocean, with reduced moisture supply towards Indochina and
southern China, in contrast to the vigorous cross-equatorial moisture-laden flow from the western Indian Ocean in the
free-running experiments. These anomalies, manifestation of a *local* aerosol effect, are indicative of the predominant role
of large-scale circulation adjustments and two-way interactions with local anomalies in realising the full extent of the
aerosol impact over Asia. Nudging the circulation outside Asia thus proves to be a strong constraint on the model response
to aerosols over Asia, despite unchanged emissions compared to the free running simulations.

In September, the dry bias over India is reduced compared to June (Fig. 5e) as in the free-running simulation. The wet
bias over China also reduces overall in magnitude compared to June, although NUDG is wetter than CONT (Fig. 1b),
which may be conducive to enhanced aerosol-cloud interactions. In fact, CDNC increases across Asia (Fig. S10c) but
cloud top effective radius decreases mainly over central and eastern China, with conversely muted changes over India
(Fig. S10d). As a result, precipitation decreases over most of central and eastern China, accompanied by positive sea-
level pressure anomalies and anomalous low-tropospheric anticyclonic circulation (Fig. 5f and 5g). The anomalous
easterly flow over northern Indochina and northern India draws anomalous southwesterly moisture transport across India,
which features widespread wetting. As during June, the lack of circulation adjustments outside Asia appears to play an
important role in determining magnitude and sign of the aerosol response: the marked anomalous anticyclone over the
subtropical western Pacific in the free-running simulations contributes to the strong southerly moisture advection toward
southern and eastern China, and thus to the generation of the precipitation increase (Fig. 3g and 3f). These features are of
very weak magnitude in the nudged experiments due to the fixed circulation, leading to prevalently dry conditions over
China (as opposed to wet anomalies). This, in turn, contributes to weakening, or even opposing, the anomalous westerly





wind across India and Indochina seen in CONT, with these regions now displaying prevalently wet (as opposed to dry)
anomalies. These findings highlight a competing role and complex interplay between sub-regional precipitation biases in
modulating the response to aerosols.

**3.4 Responses in the fixed SST PDRMIP simulations**

To ascertain whether the link between climatological biases and aerosol response found above for GA7.1 is common to
other models, we analyse the PDRMIP multi-model experiments forced by fixed SST. In particular, models are
composited based on the sign of the June precipitation bias over central India (the area 73°–85°E, 20°–28°N,
approximately corresponding to the core monsoon region), given its key role in determining the seasonality of the aerosol
imprint discussed above. Given the fundamental role of aerosol-cloud interactions in realising the aerosol impact, the
CESM1-CAM4 and GISS models are excluded from the analysis as they include only a parameterization of aerosol-
radiation interactions (Liu et al., 2018). In fact, these two models display very weak monthly precipitation variations over
India and China induced by aerosol changes (not shown). Of the five remaining models, two (i.e., HadGEM3 and IPSL-
CM) display precipitation deficit while the other three (i.e., MIROC-SPRINTARS, NorESM1and CESM1-CAM5)
present excessive rainfall over India in June (hereafter DRY and WET ensembles, respectively; Fig. S11). Biases and
responses for individual models are shown in Fig. S11.

DRY features a dipole pattern in the June precipitation bias over Asia with drying across India and most of Indochina and
wetting over China, particularly to the south and east (Fig. 6a). Based on the mechanism described above, this pattern
provides favourable conditions for aerosol-cloud interactions to come into play over China, leading to anomalous low-
tropospheric anticyclonic flow over China (Fig. S12a), thereby reducing precipitation there and shifting it southward (Fig.
6c). This leads to compensating precipitation increases over northern India, the BOB and the SCS. Key features of both
the bias and response patterns are common, in sign and magnitude, to both HadGEM3 and IPSL-CM (Fig. S11), and
overall bear marked similarly to those in HadGEM3-GA7.1 (Fig. 3). One notable difference compared to HadGEM3-
GA7.1 is that DRY shows an evident meridional land-ocean contrast in the precipitation distribution over the western
Pacific, with the wetting predominantly confined to the ocean and drier Indochina and southeastern China. This feature
is recognizable in both HadGEM3 and IPSL-CM, with the former model close to the one employed in this study, which
suggests the shift to be related to the differing prescribed SST patterns.

In order to account for model differences and to more clearly highlight the spatio-temporal changes between early and
late summer, Fig. 6b and 6f show incremental variations (i.e., September minus June differences) rather than absolute
anomalies. The DRY bias features a relative precipitation excess over India, the northern BOB, and most of Indochina,





and a deficit over eastern China. Correspondingly, the aerosol-induced response shows easterly flow (Fig. S12b) and
widespread decreased precipitation across the Indian subcontinent in September with respect to June (Fig. 6d) and,
associated with an anticyclonic anomalous flow over the SCS (Fig. S12b), contributing to precipitation increases over
southern and eastern China. There is again marked similarity between these patterns and those for the HadGEM3-GA7.1.
As noted for the June response, there is a strong land-ocean contrast in the WET precipitation distribution over the East
Asian sector.

Turning to the analysis of the WET ensemble, the June bias features precipitation excess over most of India and central
and northern China, while deficient precipitation is seen over eastern and southern China (Fig. 6e). This pattern, with
opposite anomalies over India and reversed meridional dipole over China compared to DRY, is conducive to strong
aerosol-cloud interactions over India and relatively weaker signals over eastern China (compared to DRY). As a result,
the WET response displays northeasterly flow and widespread drying over India and a cyclonic anomaly over the tropical
western Pacific leading to dry northeasterlies over central and eastern China and wet anomalies over the SCS (Fig. 6g and
S12c). The June-to-September incremental bias features an approximately opposite pattern to that in June, and so does
the precipitation response (Fig. 6f and 6h). Overall, the reversed polarity of bias and responses in WET compared to DRY
and the consistency of the key features of the patterns among the individual models further corroborate the robustness of
the physical mechanism proposed above.

## 3.5 Responses in coupled simulations

One may wonder whether the findings above, based on the analysis of atmospheric-only models, still hold in fully coupled
models and how much they are modulated by including two-way air-sea interactions. First, we analyse the PDRMIP
coupled model experiments. For consistency with the analysis of the fixed SST experiments, as well as to include the
contribution of air-sea coupling but not the full long-term response of the ocean, which presumably has not adjusted to
the time-varying emissions in the transient experiments, the analysis was restricted to the first 6–15 years of the
simulations. All five chosen models display a dry bias over India in June (Fig. S13) and thus Fig. 7 only shows the DRY
multi-model ensemble. In common with the experiments investigated above, the June bias features dry anomalies over
South Asia and wet anomalies over southern China (Fig. 7a). A noticeable difference compared to HadGEM3-GC2 is the
large-scale drying over the SCS and western subtropical Pacific, showing a meridional dipole. This dipole is obvious in
most individual models except for HadGEM3 (mostly zonal; Fig. S13). In September, the dry bias over the Indian sector
and western subtropical Pacific undergoes a marked reduction, while the wetting over China is restricted to the central
regions.



To further examine the robustness of the results, we also analyse the HadGEM3-GC2 coupled transient simulations, which
are a close counterpart to the simulations discussed in Section 3.1. The bias pattern and magnitude in the coupled
experiment bear a close similarity to that of the HadGEM3-GA7.1 model during both June and September (Fig. 8a and
8b), including the dipole between India and central-southeastern China and its sign reversal between early and late
summer. This suggests the underlying cause to be rooted in the atmospheric component (Bollasina and Nigam, 2009;
Song and Zhou, 2014). The June precipitation response features widespread wetting over India and a large southwest to
northeast oriented dipole over China, with excess precipitation over the western Pacific and drying from northern
Indochina across southeastern China to Japan (Fig. 8b). These two main features, of comparable magnitude, are also
evident in Fig. 3. The main difference is that the dipole is slightly shifted southeastward in the coupled model, associated
with the anticyclonic circulation extending over the SCS due to aerosol-induced oceanic cooling (not shown), with
consequent opposite sign precipitation anomalies over southeastern China. Also, in agreement with the atmospheric-only
simulations, the September response shows extensive drying across South Asia and wetting over southeastern China. As
in June, oceanic coupling appears to lead to some differences over southeastern China and the SCS, where the precipitation
anomalies, modulated by the slower oceanic response, are slightly shifted over the ocean compared to the uncoupled
simulations. These reversed bias patterns are similar to those noted for the PDRMIP coupled model experiments albeit
with a different orientation. Correspondingly, both the June and September responses follow the shapes of bias patterns
as those shown in Fig. 7, further attesting to the pervasiveness and consistency of the link between bias and aerosol
response across different models.
**4 Conclusions**
While numerous studies have emphasised the key role of anthropogenic aerosols in driving seasonal-mean changes in the
Asian monsoon, only very few of them have focused on the aerosol impact at sub-seasonal (e.g., monthly) time scale (e.g.,
Lau and Kim, 2006; Bollasina et al., 2013; Fang et al., 2023). Yet, the onset and withdrawal phases of the monsoon are
of key importance for the regional economy and water resources as they herald the arrival and demise of the monsoon
rains, which provide up to 75% of the total annual rainfall for large areas of Asia. For example, a delayed monsoon onset
as well as an early monsoon retreat, or long-term trends in their timings induced by anthropogenic aerosols, can lead to
severe consequences for the region. Equally important, inter-model discrepancies in the simulated aerosol-induced
monsoon changes at sub-seasonal scale may help to explain the diversity of the seasonal-mean responses.

Based on the analysis of several climate models and aerosol forcing experiments, we find the sub-seasonal variability of
the Asian summer monsoon response to regional anthropogenic aerosol changes to be significantly affected by the spatial
pattern and seasonality of the model bias across Asia. The aerosol impact on monsoon precipitation and circulation is



strongly influenced by the model ability to simulate the spatial distribution and temporal variability of the climatological
monsoon clouds and precipitation, as well as the underlying atmospheric dynamical action centres. These critically
modulate the magnitude and efficacy of aerosol-cloud-precipitation interactions, which are the predominant driver of the
total aerosol response (e.g., Li et al., 2018; Dong et al., 2021). The amount of available water vapour in the model baseline
climatological state exerts a strong control on the extent to which aerosols can interact with clouds and precipitation
processes (i.e., via reduced cloud effective radius) and thus modulate the aerosol-induced monsoon response. This
involves a strong interplay between South and East Asia and their relative predominance in driving the overall monsoon
response, with a striking contrast between the early and late summer aerosol-driven changes ascribable to the seasonal
evolution of the biases between the two regions. Our results and proposed mechanism, firstly based on a detailed analysis
of atmospheric-only experiments with the HadGEM3-GA7.1 model, are corroborated by the analysis of other atmospheric
and coupled models for which sensitivity experiments to Asian aerosol changes are also available.

In summary, during the onset month (June), models that feature a dry bias over India also display corresponding wet
anomalies over eastern China. As the monsoon season progresses and approaches the end (September), the absolute bias
decreases, or even reverses, such that incremental changes show wetter conditions over India and drier over eastern China
compared to June. Similar variations, but of opposite sign, occur for the models that display a wet June bias over India
(and corresponding deficient rainfall over China). These patterns and their sub-seasonal evolution, together with the
corresponding atmospheric circulation anomalies, indicate the existence of a strong internal coupling between the South
and East Asian monsoon systems, whereby the two components fluctuate and oppose each other at short (monthly or
below) time scales. As a result, the aerosol influence on the monsoon, driven by the magnitude of aerosol-cloud
interactions, also features a dipole and oscillating pattern between South and East Asia, with the key driving region
varying during the season and depending on the evolution of the model climatological state. For example, while the direct
aerosol imprint is predominant over East Asia in early summer, it is dominating over South Asia towards the end of the
season in the DRY composites. The continental-scale aerosol response, particularly the inter-monsoon interaction,
involves an ensuing large-scale atmospheric circulation response, which is pivotal to extending the aerosol impact
downstream of the dominating aerosol-forcing region by modulation of the associated moisture transport towards the rest
of the domain. The analysis of the nudged experiments further supports the crucial role of non-regional atmospheric
circulation adjustments: while keeping the circulation outside Asia close to observations reduces the model bias over Asia,
the lack of adjustments under varying Asian aerosol emissions dampens and modifies the pattern and evolution of the
regional precipitation response, leading to unrealistic changes (e.g., seasonal mean wetting over South Asia). This
suggests that climatological large-scale circulation features, such as the western Pacific subtropical high and the
Mascarene high over the southern Indian Ocean, are not only modulated by aerosol forcing over Asia but are also active
contributors to generating the aerosol impact itself over Asia.




The consistency of our findings across different models suggests that the mechanism is robust with respect to the specific
model structure and physics, including details of the aerosol module, as long as aerosol-cloud interactions are
parameterized. Biases and responses are markedly similar between atmosphere-only and coupled models over land (e.g.,
South and East Asia), where the largest aerosol loading is also located and thus the largest forcing is exerted via aerosol-
cloud-precipitation interactions. Response patterns between uncoupled and coupled models differ in both magnitude and
sign over the surrounding oceanic regions. However, the coupled model response pattern displays an overall minor
sensitivity to changes in the averaging period (see Fig. S14 and Fig. 7), with the key anomalies, particularly over land,
appearing already in the first decades of the simulation. This indicates that, while air-sea interactions contribute to
realising the aerosol impact, the full oceanic response plays a secondary role compared to the predominant action of the
atmospheric circulation (Soden and Chung, 2017). This topic, and particularly the analysis of the time scales required to
set-up the equilibrium response, has been mostly overlooked in literature, which often compared the fast to the slow
response, the latter taken after 50 or more simulated years (e.g., Samset et al., 2016).

One important implication of the link between model climatological bias and response pattern found here is the possibility
of better understanding and constraining the diversity and inconsistencies of model responses to aerosol changes over
Asia in historical and future projections by accounting for model deficiencies in simulating the climatological monsoon
seasonal cycle compared to observations. This will help in further narrowing the uncertainties associated with aerosol-
cloud interactions, given their predominant role in driving the monsoon changes. For example, the clear contrast in the
monthly response to aerosols between the PDRMIP DRY and WET composites calls for caution in the interpretation of
the aerosol-induced signal without proper consideration of the model baseline performance. This will translate into more
robust assessments of sub-regional scale monsoon variations. In fact, despite an overall similarity in the seasonal mean
monsoon responses between the DRY and WET ensembles, it is interesting to notice that the difference pattern in
precipitation (e.g., DRY minus WET) bears a striking similarity to the observational pattern (Fig. S15 and Fig. S1).

Differences exist between the observed monsoon changes during the recent decades shown above (Jin and Wang, 2017;
Monerie et al., 2022) and those over a longer period (e.g., late 20[th] century) documented in previous literature and
attributed to the dominating regional aerosol forcing, and sulfate aerosols in particular. For example, the increase in
anthropogenic aerosol emissions over Asia in the second half of the 20[th] century has been found to play a key role in
driving the observed southern flood-northern dry rainfall dipole over East Asia (Gong and Ho, 2002; Song et al., 2014b;
Dong et al., 2016), as well as the South Asian monsoon decline (Gu et al., 2006; Bartlett et al., 2018; Dong et al., 2016;
Jiang et al., 2013). A combination of factors, including internal climate variability, may have contributed to these recent
trends (Huang et al., 2020b; Monerie et al., 2022) or to set background (oceanic) conditions on top of which aerosols



acted (Lin et al., 2016b). While the findings of our study on the possible role of aerosols are necessarily not conclusive,
the model bias is found to be equally important to explain model discrepancies. A careful examination of these biases
could help reconcile the generally poor performance of state-of-the-art climate models in reproducing recently observed
trends (Huang et al., 2020b; Monerie et al., 2022).

It is also worth emphasizing that the analysis carried out above focuses on the impact of sulfate aerosol emission changes,
either because of their marked dominance over other aerosol components (e.g., BC and OC) in the historical period
investigated with the HadGEM3-GA7.1 and HadGEM3-GC2 experiments or because of experimental design in the
PDRMIP simulations. While sulfate aerosol emissions underwent the largest changes across Asia throughout the historical
period (e.g., Lund et al., 2019), the imprint of BC aerosols, although of comparatively weaker magnitude (e.g., Liu et al.,
2018; Westervelt et al., 2018), needs also to be accounted to interpret the full extent of the simulated monsoon response
to historical aerosol changes and its inter-model inconsistencies given, for example, their different physical mechanisms
and responses of opposite sign compared to those due to sulfate aerosols (e.g., Xie at al., 2020).

The competition between South Asia and East Asia in generating the continental-scale monsoon response and the
underpinning modulation by the bias pattern is very relevant in the context of interpreting near-future monsoon projections
and related uncertainties, including for regional attribution studies, given the present-day and near-future dipole pattern
of emission changes between the two regions (Lund et al., 2019; Samset et al., 2019). For example, it is conceivable to
expect that a reduced model bias over South Asia, particularly in early summer, would further promote the importance of
Indian aerosol emissions compared to those over China. This also highlights the potential key role of non-local aerosols
in driving the simulated response across Asia, which is again crucial in interpreting future projections.

We acknowledge some limitations of this study. Only a few models are available in each of the DRY and WET composites
as aerosol-cloud interactions are not parameterised in some of the PDRMIP models. There are also inter-model differences
in the aerosol setups (i.e., prescribed concentrations or emissions) the implications of which are difficult to ascertain given
the limited model sample. Including more models and conducting coordinated perturbed aerosol experiments to Asian
aerosols would further increase the robustness of our study. It would be interesting to extend this analysis to a longer
period and examine, for example, the 20th-century monsoon changes. Internal climate variability may also play an
important role and partially mask or offset externally-driven changes, especially given the relatively short time period
examined here.

**Data availability**. The GPCP and CMAP observational datasets are obtained from
https://www.esrl.noaa.gov/psd/data/grid-ded/data.gpcp.html and https://psl.noaa.gov/data/gridded/data.cmap.html,



respectively. The ERA-I reanalysis used for nudging can be accessed from
https://www.ecmwf.int/en/forecasts/datasets/reanalysis-datasets/era-interim (Dee et al., 2011). The ERA5 reanalysis is
provided by the European Center for Medium-Range Weather Forecasts
(https://www.ecmwf.int/en/forecasts/dataset/ecmwf-reanalysis-v5) (Hersbach et al., 2020). The PDRMIP data can be
accessed through the World Data Center
for Climate (WDCC) data server at https://doi.org/10.26050/WDCC/PDRMIP_2012-2021 (Andrews et al., 2021). The
model simulation output is available from the corresponding author on reasonable request.

**Author contribution.** MAB and ZL designed the study. ZL ran the model simulations. ZL and MAB carried out the
analysis, visualized the results and discussed the results. All authors edited the paper.

**Competing interests.** At least one of the (co-)authors is a member of the editorial board of Atmospheric Chemistry and
Physics. The authors also have no other competing interests to declare.

**Acknowledgements**. ZL is supported by the start-up funding (G0101000155) of the Hong Kong University of Science
and Technology (Guangzhou). MB is supported by the Natural Environment Research Council (grant no. NE/N006038/1).
MB and LW acknowledge support from the Research Council of Norway (grant no. 324182; CATHY). ZL, MB, and LW
were supported by the UK-China Research and Innovation Partnership Fund through the Met Office Climate Science for
Service Partnership (CSSP) China as part of the Newton Fund. We acknowledge the use of ARCHER, the UK HPC, and
of the JASMIN super-data-cluster.

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



Table 1. Model simulations carried out in this study. The Asia domain (10°–45°N, 60°–125°E) is enclosed by the purple box in Fig.
1c. Note wind nudging is applied only above the planetary boundary layer (model level 12, or approximately 850 hPa). Years 2003 –
2012 are used for analysis.

| Experiment | Description |
| --- | --- |
| CONT | Transient Asian aerosols during 1991–2012 and without nudging |
| CONTfA | Asian aerosols fixed at their 1991 values and without nudging |
| NUDG | Same as CONT except for wind nudging outside Asia |
| NUDGfA | Same as CONTfA except for wind nudging outside Asia |






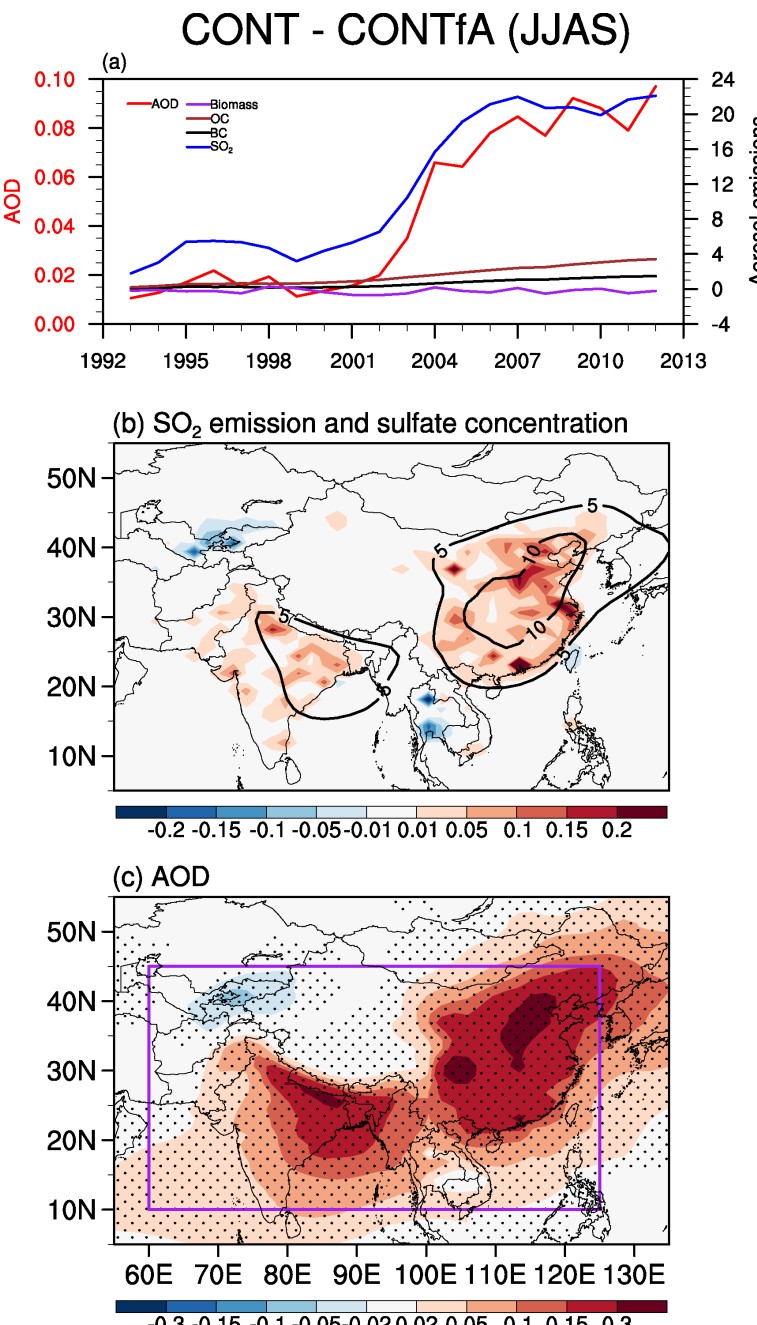

Fig. 1. (a) Differences of annual time series of summer AOD (unitless; red), total $SO_2$ emissions (Tg yr$^{-1}$; blue), total BC emissions (Tg yr$^{-1}$; black), total OC emissions (Tg yr$^{-1}$; brown), and total biomass burning emissions (Tg yr$^{-1}$; purple) over Asia between CONT and CONTfA. Spatial distribution of (b) $SO_2$ emissions (shading; Tg yr$^{-1}$) and sulfate column burden (contour; mg m$^{-2}$) and (c) AOD changes (difference between CONT and CONTfA averaged for the period 2003–2012). The purple box in (c) denotes the Asia region (10°–45°N, 60°–125°E). Black dots in (c) mark grid-points for which the difference is significant at the 90% confidence level.



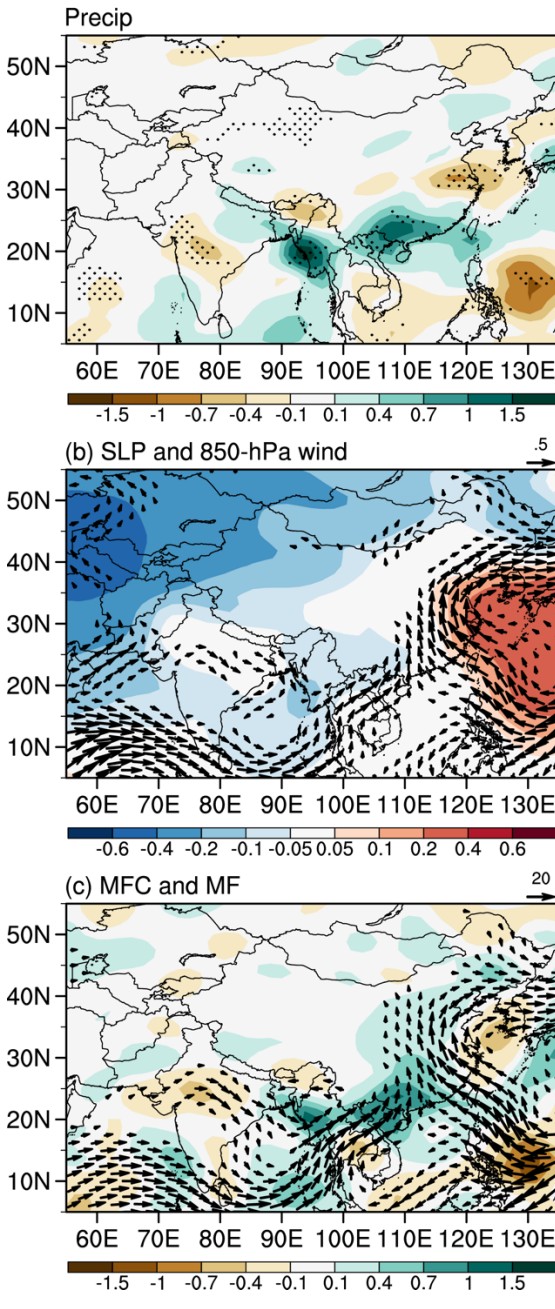


Fig. 2. JJAS response to Asian anthropogenic aerosols (difference between CONT and CONTfA averaged during 2003-2012) for (a)
precipitation (mm day$^{-1}$), (b) sea-level pressure (hPa; shades) and 850-hPa winds (m s$^{-1}$), and (c) 1000–300 hPa vertically integrated
moisture flux convergence (mm day$^{-1}$, shades) and moisture flux (kg m$^{-1}$ s$^{-1}$). Black dots in (a) mark grid-points for which the difference
is significant at the 90% confidence level.

Fig. 3. (a) June precipitation bias (mm day$^{-1}$) in CONT with respect to the mean of GPCP and CMAP. Model data are averaged over 2003–2012, observations over 1981–2010. June response to Asian anthropogenic aerosols (difference between CONT and CONTfA averaged during 2003-2012) for (b) precipitation (mm day$^{-1}$), (c) sea-level pressure (hPa, shades) and 850-hPa wind (m s$^{-1}$), and (d) 1000–300 hPa vertically integrated moisture flux convergence (mm day$^{-1}$, shades) and moisture flux (kg m$^{-1}$ s$^{-1}$). (e–h) Same as (a–d) but for September. Black dots in (b) and (f) mark grid-points for which the difference is significant at the 90% confidence level.





Fig. 4. June response to Asian anthropogenic aerosols (difference between CONT and CONTfA averaged during 2003-2012) for (a) cloud droplet number concentration ($10^{10}$ m$^{-2}$), (b) cloud-top effective radius (μm), (c) liquid water path (g m$^{-2}$), and (d) cloud fraction (%). (e–h) Same as (a–d) but for September. Black dots mark grid-points for which the difference is significant at the 90% confidence level.



923
924    Fig. 5. Same as Fig. 3 but for the difference between the corresponding nudged simulations (i.e., NUDG–NUDGfA).





925
Fig. 6. DRY PDRMIP model composites in (a) June precipitation bias (mm day$^{-1}$), (b) September minus June difference in precipitation
bias, (c) June precipitation response to increased Asian sulfate aerosols (i.e., the difference between 10× sulfate and baseline
simulations), and (d) September minus June difference in the precipitation response to increased Asian sulfate aerosols. (e–h) Same as
(a–d) but for WET PDRMIP model composites. Black dots mark grid-points for which all models agree on the sign of the precipitation
differences.

## Precip bias

(a) Jun

(b) Sep-Jun

## Precip

(c) Jun

(d) Sep-Jun

Fig. 7. PDRMIP coupled model composites in (a) June precipitation bias (mm day$^{-1}$), and (b) June precipitation response to increased Asian sulfate aerosols (i.e., the difference between 10× sulfate and baseline simulations). (c)-(d): Same as (a)-(b) but for the September minus June differences. Black dots mark grid-points for which at least four out of the five models agree on the sign of the precipitation differences.



Fig. 8. (a) June precipitation bias (mm day$^{-1}$) in HadGEM3-GC2 coupled simulations, (b) precipitation bias difference between September and June, (c) June precipitation response to Asian aerosol changes, (d) difference in the precipitation response to Asian aerosols between September and June. Black dots in (b) and (d) mark grid-points for which the difference is significant at the 90% confidence level.