# Peer review of "Impact of Asian aerosols on the summer monsoon strongly modulated by regional precipitation biases"

_EGUsphere, 2023_

## Referee Comment (RC1)

**General comments:**

This article titled "Impact of Asian aerosols on the summer monsoon strongly modulated by regional precipitation biases" mainly discusses the challenges of the Asian summer monsoon to climate models, as well as the mutual influence between model bias and atmospheric circulation. However, can some updated data be provided in this manuscript?

**Specific comments:**

Some images have a poor appearance, such as Figure 3 (g), and the arrows can be adjusted to be thinner. The colorbar can be further refined or a smooth one can be used, as many details cannot be displayed under the current colorbar.

The dataset used in the article seems to lack a quantifiable validation of its accuracy. A quantifiable validation is needed to evaluate its accuracy.

The selection of parameters is usually a crucial step in model development and use, and the article seems to lack detailed explanation of the model's parameter settings.

**Technical comments:**

**Line 53:** "could albedo" → "cloud albedo"

**Line 53:** " "cloud albedo and lifetime, and precipitation processes"→ "cloud albedo, lifetime, and precipitation processes" There are other errors like this in the text, please check carefully

**Line 67:** "South and East Asian aerosols separately exert a strong influence"→"South and East Asian aerosols exert a strong influence separately"

**Line 72:** "the Asian monsoon march" There is a spelling error or misuse of vocabulary here.

**Line 141:** "in coupled mode ((Liu et al., 2018)." →" in coupled mode (Liu et al., 2018)."

**Line 218:** "Inspection of monthly precipitation and low-level circulation changes reveals a stark contrast over the Indian subcontinent and adjacent ocean between the early and late monsoon season: increased precipitation and anomalous cyclonic flow over the BOB in June, consistent with the seasonal mean, and decreased precipitation

and anomalous anticyclonic winds over India in September (Figs. S2 and S3)." This sentence may be too long, consider splitting it into two or more concise sentences.

**Line 241:** "The accuracy of the simulated regional climate change signal and its attribution to anthropogenic drivers have been suggested to be strongly dependent on the model performance in reproducing the corresponding mean climatological conditions, which represent the baseline state on top of which changes occur (Matsueda and Palmer, 2011; Christidis et al., 2013)." → "has been"

**Line 452:** "For consistency with the analysis of the fixed SST experiments" → "For consistency with the analysis of the experiments with fixed SST"

---

## Author Comment (AC1)

Response to Review Comments

Dear Editor,

We thank you and both reviewers very much for their careful review and valuable comments on our manuscript. We have tried our best to address all concerns and revised the manuscript accordingly. Please note that the reviewer's remarks are in black, our response is highlighted in blue, and extracts from the manuscript are in red, with **new**

**texts** that have been added/edited marked in bold. We hope that you find revised manuscript satisfactory. Thank you very much.

Kind regards,

Zhen LIU, on behalf of all co-authors

**Responses to Reviewer #1:**

**General comments:**

This article titled "Impact of Asian aerosols on the summer monsoon strongly modulated by regional precipitation biases" mainly discusses the challenges of the

Asian summer monsoon to climate models, as well as the mutual influence between model bias and atmospheric circulation. However, can some updated data be provided in this manuscript?

*Response*: Thank you for the comments and suggestions. A point-by-point response is given below.

**Specific comments:**

1. Some images have a poor appearance, such as Figure 3 (g), and the arrows can be adjusted to be thinner. The colorbar can be further refined or a smooth one can be used, as many details cannot be displayed under the current colorbar.

*Response*: Thanks for your suggestions. The arrows are thinner for all vector plots in the revised manuscript. To keep consistency of all figures, we have carefully adjusted the colorbar scale, which considerably improves the readability of the plot. The figure below is the new Fig. 3.

[Figure]

Fig. 3. (a) June precipitation bias (mm day$^{-1}$) in CONT with respect to the mean of GPCP and CMAP. Model data are averaged over 2003–2012, observations over 1981–2010. June response to Asian anthropogenic aerosols (difference between CONT and CONTfA averaged during 2003-2012) for (b) precipitation (mm day$^{-1}$), (c) sea-level pressure (hPa, shades) and 850-hPa wind (m s$^{-1}$), and (d) 1000–300 hPa vertically integrated moisture flux convergence (mm day$^{-1}$, shades) and moisture flux (kg m$^{-1}$ s$^{-1}$). (e–h) Same as (a–d) but for September. Black dots in (b) and (f) mark grid-points for which the difference is significant at the 90% confidence level.

2. The dataset used in the article seems to lack a quantifiable validation of its accuracy. A quantifiable validation is needed to evaluate its accuracy.

*Response*: Thanks for your suggestions. To provide a basic evaluation of the model performance in simulating the key features of the Asian summer monsoon, Figure 1

compares the 1993-2012 June–September average precipitation and 850-hPa winds in the control simulation to observations (GPCP and CMAP average for precipitation,

ERA5 for wind). The model reproduces the broad characteristics of the observed rainfall and circulation patterns (pattern correlation of 0.80 for precipitation, which is significant at the 99.9% confidence level). The difference panel indicates that the model is too dry over India due to a weaker southwesterly monsoon flow, but features wet anomalies over southwestern China and the northwestern subtropical Pacific associated with enhanced cyclonic flow. Note that this bias pattern is common across CMIP6

models, although the magnitude of the anomalies vary from model to model (Wilcox et al., 2020), and is also consistent with that in the historical simulations of the CMIP6

Met Office model (Rajendran et al., 2022). A thorough discussion of the model bias and its linkage to regional and remote circulation are documented in Liu et al. (2021).

We have integrated this figure and related description in the main text as follows:

[Figure]

Figure 1. June–September average precipitation (mm day$^{-1}$) and 850-hPa wind (m s$^{-1}$) for the observations (GPCP
and CMAP average for precipitation, ERA5 for wind), the control simulation, and their differences (model
simulations minus observations) during the period 1993 to 2012.

Lines 179–187: "**Figure 1 compares the 1993-2012 June–September average**

**precipitation and 850-hPa winds in the control simulation to observations (GPCP**

**and CMAP average for precipitation, ERA5 for wind). The model reproduces the**

**broad characteristics of the observed rainfall and circulation patterns (pattern**

**correlation of 0.80 for precipitation, which is significant at the 99.9% confidence**

**level). The difference panel indicates that the model is too dry over India due to a weaker southwesterly monsoon flow, but features wet anomalies over southwestern China and the northwestern subtropical Pacific associated with enhanced cyclonic flow. Note that this bias pattern is common across CMIP6 models, although the magnitude of the anomalies varies from model to model (Wilcox et al., 2020), and is also consistent with that in the historical simulations of the CMIP6 Met Office model (Rajendran et al., 2022). A thorough discussion of the model bias and its linkage to regional and remote circulation can be found in Liu et al. (2021).”**

3.  The selection of parameters is usually a crucial step in model development and use, and the article seems to lack detailed explanation of the model's parameter settings.

*Response*: Thanks for your suggestions. We have provided more details on the selection of the model parameters.

Lines 104–110: “GA7.1 was used as the atmospheric component of the climate model participating in CMIP6, **which reduces the overly negative global-mean anthropogenic aerosol effective radiative forcing in the previous model version, GA7.0 (Walters et al., 2019). A single-moment microphysics is used based on Wilson and Ballard (1999), with extensive improvement of the warm rain scheme (Boutle et al., 2014a, b). To account for aerosol-cloud interactions, the cloud droplet number concentration is calculated using prognostic aerosol concentration according to the UK Chemistry and Aerosol (UKCA)-Activate scheme (West et al., 2014). The atmospheric boundary layer and convection schemes are based on Lock et al. (2000) and Gregory and Rowntree (1990), respectively. A detailed description of the HadGEM3-GA7.1 physics is provided by Walters et al. (2019).”**

Technical comments:

4.  Line 53: “could albedo” → “cloud albedo”

*Response*: Changed.

5.    Line 53: "cloud albedo and lifetime, and precipitation processes"→ "cloud albedo, lifetime, and precipitation processes" There are other errors like this in the text, please check carefully

*Response*: Thank you for spotting the error. We have gone through the whole manuscript carefully and revised it accordingly.

6.    Line 67: "South and East Asian aerosols separately exert a strong influence"→"South and East Asian aerosols exert a strong influence separately"

*Response*: Thank you for your comment. Here, we are trying to say that either South or

East Asian aerosols can affect both the South and East Asian monsoons. Sorry for the confusion. We revised the sentence as follows:

Lines 67–69: "In particular, **either** South **or** East Asian aerosols **can** exert a strong influence on both the South and East Asian monsoons, with contrasting, if not opposite, changes as well as strong non-linear interactions between the responses to individual emission sources."

7.    Line 72: "the Asian monsoon march" There is a spelling error or misuse of vocabulary here.

*Response*: We revise the word "march" to "progression".

8.    Line 141: "in coupled mode ((Liu et al., 2018)." →" in coupled mode (Liu et al.,

2018)."

*Response*: Done.

9.    Line 218: "Inspection of monthly precipitation and low-level circulation changes reveals a stark contrast over the Indian subcontinent and adjacent ocean between the early and late monsoon season: increased precipitation and anomalous cyclonic flow over the BOB in June, consistent with the seasonal mean, and decreased precipitation and anomalous anticyclonic winds over India in September (Figs. S2 and S3)." This sentence may be too long, consider splitting it into two or more concise sentences.

*Response*: Thanks for your suggestions. We split it into three sentences:

Lines 239–242: "Inspection of monthly precipitation and low-level circulation changes reveals a stark contrast over the Indian subcontinent and adjacent ocean between the early and late monsoon season (Figs. S2 and S3). In June, **there is** increased precipitation and anomalous cyclonic flow over the BOB, consistent with the seasonal mean. **On the contrary**, decreased precipitation and anomalous anticyclonic winds **are**

**seen** over India in September."

10. Line 241: "The accuracy of the simulated regional climate change signal and its attribution to anthropogenic drivers have been suggested to be strongly dependent on the model performance in reproducing the corresponding mean climatological conditions, which represent the baseline state on top of which changes occur (Matsueda and Palmer, 2011; Christidis et al., 2013)." → "has been"

*Response*: Per your suggestions.

11. Line 452: "For consistency with the analysis of the fixed SST experiments" →

"For consistency with the analysis of the experiments with fixed SST"

*Response*: Per your suggestions.

**References**

Boutle, I. A., Eyre, J. E. J., and Lock, A. P.: Seamless Stratocumulus Simulation across the Turbulent Gray Zone, Monthly Weather Review, 142, 1655–1668, https://doi.org/10.1175/MWR-D-13-00229.1, 2014a.

Boutle, I. A., Abel, S. J., Hill, P. G., and Morcrette, C. J.: Spatial variability of liquid cloud and rain: observations and microphysical effects, Quarterly Journal of the Royal Meteorological Society, 140, 583–594, https://doi.org/10.1002/qj.2140, 2014b.

Gregory, D. and Rowntree, P. R.: A Mass Flux Convection Scheme with Representation of Cloud Ensemble Characteristics and Stability-Dependent Closure, Monthly Weather Review, 118, 1483–1506, https://doi.org/10.1175/1520-0493(1990)118<1483:AMFCSW>2.0.CO;2, 1990.

Liu, Z., Bollasina, M. A., Wilcox, L. J., Rodríguez, J. M., and Regayre, L. A.: Contrasting the Role of Regional and Remote Circulation in Driving Asian Monsoon Biases in MetUM GA7.1, Journal of Geophysical Research: Atmospheres, 126, https://doi.org/10.1029/2020JD034342, 2021.

Lock, A. P., Brown, A. R., Bush, M. R., Martin, G. M., and Smith, R. N. B.: A New Boundary Layer Mixing Scheme. Part I: Scheme Description and Single-Column Model Tests, Monthly Weather Review, 128, 3187–3199, https://doi.org/10.1175/1520-0493(2000)128<3187:ANBLMS>2.0.CO;2, 2000.

Rajendran, K., Surendran, S., Varghese, S. J., and Sathyanath, A.: Simulation of Indian summer monsoon rainfall, interannual variability and teleconnections: evaluation of CMIP6 models, Climate Dynamics, https://doi.org/10.1007/s00382-021-06027-w, 2022.

Walters, D., Baran, A. J., Boutle, I., Brooks, M., Earnshaw, P., Edwards, J., Furtado, K., Hill, P., Lock, A., Manners, J., Morcrette, C., Mulcahy, J., Sanchez, C., Smith, C., Stratton, R., Tennant, W., Tomassini, L., Van Weverberg, K., Vosper, S., Willett, M., Browse, J., Bushell, A., Carslaw, K., Dalvi, M., Essery, R., Gedney, N., Hardiman, S., Johnson, B., Johnson, C., Jones, A., Jones, C., Mann, G., Milton, S., Rumbold, H., Sellar, A., Ujiie, M., Whitall, M., Williams, K., and Zerroukat, M.: The Met Office Unified Model Global Atmosphere 7.0/7.1 and JULES Global Land 7.0 configurations, Geoscientific Model Development, 12, 1909–1963, https://doi.org/10.5194/gmd-12-1909-2019, 2019.

West, R. E. L., Stier, P., Jones, A., Johnson, C. E., Mann, G. W., Bellouin, N., Partridge, D. G., and Kipling, Z.: The importance of vertical velocity variability for estimates of the indirect aerosol effects, Atmospheric Chemistry and Physics, 14, 6369–6393, https://doi.org/10.5194/acp-14-6369-2014, 2014.

Wilcox, L. J., Liu, Z., Samset, B. H., Hawkins, E., Lund, M. T., Nordling, K., Undorf, S., Bollasina, M., Ekman, A. M. L., Krishnan, S., Merikanto, J., and Turner, A. G.: Accelerated increases in global and Asian summer monsoon precipitation from future aerosol reductions, Atmospheric Chemistry and Physics, https://doi.org/10.5194/acp-20-11955-2020, 2020.

Wilson, D. R. and Ballard, S. P.: A microphysically based precipitation scheme for the UK
meteorological office unified model, Quarterly Journal of the Royal Meteorological Society, 125, 1607–
1636, https://doi.org/10.1002/qj.49712555707, 1999.

---

## Author Comment (AC2)

Response to Review Comments

Dear Editor,

We thank you and both reviewers very much for their careful review and valuable comments on our manuscript. We have tried our best to address all concerns and revised the manuscript accordingly. Please note that the reviewer's remarks are in black, our response is highlighted in blue, and extracts from the manuscript are in red, with **new**

**texts** that have been added/edited marked in bold. We hope that you find revised manuscript satisfactory. Thank you very much.

Kind regards,

Zhen LIU, on behalf of all co-authors

**Responses to Reviewer #2:**

This study examines the link between monsoon biases relative to observations and monsoon response to anthropogenic aerosols in Asia in terms of monsoonal precipitation, circulations, moisture budget using numerical experiments. The paper tries to address an important question: how do modelled precipitation biases influence anthropogenic aerosol-induced monsoon changes. Overall, it is an interesting paper with detailed analysis. At the same time, it is a very long paper: 8 figures in the main text plus 15 figures in the supplementary materials. The authors should include as many figures as possible in the main text rather than in the supplementary. The figures are not clearly labelled (some figure captions are misleading); some figures in the supplementary materials can be combined with the figures in the main text. I suggest the authors include all simulations/experiments in Table 1 with clear description. The result part contains too much discussion of previous studies, which significantly distract the audience's attention. The discussion can be replaced to a new Discussion section close to the end of the paper. Moreover, the sections in the Result 3.1 and 3.2 now are too long and may be divided into subsections. Overall, it is hard to follow the entire paper (I have to often refer to the supplementary figures). I hope by reorganizing the result sections, redesigning some of the figures, correcting figure captions, the authors could improve the quality of the manuscript in a significant way to meet the standards of ACP.

*Response*: Thank you for the comments and suggestions. A point-by-point response is given below. In particular:

1. We have reconsidered the figures set and we have moved several of them from the supplementary material to the main text. There are now 14 and 8 figures in the main manuscript and supplementary file, respectively.

2. We have corrected the figure titles and captions, which hopefully makes the figure clearer.

3. All the experiments used in this study are included in Table 1.

43  4. We have moved the discussion in the result part to a new discussion section

44   before the summary and conclusions part.

45  5. Results 3.1 and 3.2 have been split into subsections.

**Major comments**

47 In several places, the authors mentioned that aerosol–cloud interactions dominate the

48 aerosol-induced monsoonal changes, for example, Line 512. In my understanding,

49 aerosol–radiation interactions also play an important role in modulating monsoon

50 rainfall, sometime even a bigger role than aerosol–cloud interactions. I saw the authors

51 analyzed the cloud responses to anthropogenic aerosols. However, without a direct

52 comparison of monsoonal precipitation responses to aerosol–cloud interactions and

53 aerosol–radiation interactions, the authors should be careful with their wording. I am

54 wondering if the authors could separate the two interactions in their analysis/model,

55 which would provide very interesting analysis and results and improve the scientific

56 implication of this paper.

57 *Response*: Thanks for pointing this out. Unfortunately, we cannot separate the two

58 interactions without additional experiments.

59 Given the limited space for this paper, we have replaced the words of "driven",

60 "predominant" with the word "important", "modulated", and "key" in the revised text.

61 In the response to the specific comment #13, we briefly discussed the relative

62 importance of aerosol-radiation interactions and aerosol-cloud interactions.

63 Lines 23–26: "The aerosol impact on monsoon precipitation and circulation is strongly

64 influenced by a model's ability to simulate the spatiotemporal variability of the

65 climatological monsoon winds, clouds and precipitation across Asia, which modulates

66 the magnitude and efficacy of aerosol-cloud-precipitation interactions, an **important**

67 **component** of the total aerosol response."

68 Lines 422–424: "Given the **key** role of aerosol-cloud interactions in realising the

69 aerosol impact, the CESM1-CAM4 and GISS models are excluded from the analysis as

70 they include only a parameterization of aerosol-radiation interactions (Liu et al., 2018)."

Lines 515–516: "These biases critically modulate the magnitude and efficacy of aerosol-cloud-precipitation interactions, **an important component** of the total aerosol-**driven** response."

Lines 517–518: "This will help in further narrowing the uncertainties associated with aerosol-cloud interactions, given their **important** role in driving the monsoon changes."

Lines 586–588: "As a result, the aerosol influence on the monsoon, **modulated** by aerosol-cloud interactions, also features a dipole and oscillating pattern between South and East Asia, with the key driving region varying during the season, and depending on the evolution of the model climatological state."

1. Line 47: It is not clear what trends are driven by aerosols?

*Response*: Here we mean temperature and precipitation. We revise the sentence as follows:

Lines 45–48: "In particular, model biases introduce large uncertainties in our ability to separate externally-forced from internally-generated monsoon variability, preventing robust attribution to specific drivers, including the extent to which recent and near-future trends of **temperature and precipitation over East Asia** are driven by anthropogenic aerosols (Wilcox et al., 2015; Dai et al., 2022)"

2. Line 98: What is the GLOMAP scheme? Spell out its full name.

*Response*: Thanks for your comments. GLOMAP is short for Global Model of Aerosol Processes. We have revised the sentence in the manuscript accordingly.

3. 1b: Caption is not clear: why emissions can be negative, should be emission differences.

*Response*: Thanks for spotting out this error. We have corrected it.

4. Line 181–182: Northern India should be deleted because precipitation increases is not statistically significant.

*Response*: Per your suggestions.

5.  2b–2c: grid cells with statistically significant changes represented by shadings should be highlighted as in Fig. 2a.

*Response*: Per your suggestions. We have highlighted the significant changes in other main figures as well.

6.  Lines 183–184: "The simultaneous northwestward shift and strengthening of the

Mascarene High over the equatorial Indian" is not shown in Fig. 2. The white colors represent close-to-zero changes in SLP.

*Response*: Thank you for pointing this out. There are positive sea-level pressure anomalies over the region, 20°S–20°N, 25°–60°E (Figure R1b, reported below), indicating a northwestward shift and strengthening of the Mascarene High over the equatorial Indian. To keep the consistency of the focused domain, we have removed this argument to avoid confusion.

[Figure]

Figure R1. JJAS response to Asian anthropogenic aerosols (difference between CONT and CONTfA averaged during 2003–2012) for (a) precipitation (mm day$^{-1}$), (b) sea-level pressure (hPa; shades) and 850-hPa winds (m s$^{-1}$), and (c) 1000–300 hPa vertically integrated moisture flux convergence (mm day$^{-1}$, shades) and moisture flux (kg m$^{-}$
$^1$ s$^{-1}$). Black dots mark grid-points for which the difference is significant at the 90% confidence level.

7.   What's the difference between Fig. 1 and Fig. S9?

*Response*: Fig. 1 shows the differences between CONT and CONTfA, while Fig. S9

shows the differences between NUDG and NUDGfA, the pair of experiments in which the large-scale circulation outside Asia is nudged toward ERA-I reanalysis. Comparing the differences between the free-running experiments (i.e., CONT – CONTfA) and the nudged runs (i.e., NUDG – NUDGfA) enable us to determine the extent to which simultaneous adjustments in the large-scale atmospheric circulation outside the region modulate the Asian monsoon response to changes in regional anthropogenic aerosols.

The AOD changes are similar between Fig. 1 and Fig. 9 although circulation and precipitation differences are distinct, suggesting that the AOD changes are mainly driven by emission changes rather than aerosol transport and removal processes.

8.   S1 can be combined with Fig. 2 with 3 rows and 2 columns.

*Response*: Per your suggestions.

9.   Line 201: should be "aerosol-driven rainfall difference pattern."

*Response*: Thanks for your comment. Corrected.

10. 3a: Why not use the same period for model and observations: 2003–2012?

Monsoon precipitation shows strong interannual and decadal variations, which should be considered when comparing model and observations.

*Response*: We agree that there are interannual and decadal variations. However, the bias is normally estimated relative to a long-term climatology, and the present-day climatology is commonly calculated based on a 30-year period from 1981 to 2010. We also examine the June and September biases relative to observation over 2003–2012

(Figure R2c and R2d below). The patterns are very similar to those using observations over 1981–2010, suggesting that our results are not sensitive to the choice of the climatological period. As such, we will keep using the period 1981–2010 to calculate the climatology and subsequent model biases.

[Figure]

Figure R2. (a) June and (b) September precipitation bias (mm day$^{-1}$) in CONT with respect to the mean of GPCP
and CMAP. Model data is averaged over 2003–2012, observations are averaged over 1981–2010. (c) and (d) Same
as (a) and (b) but observations are averaged over 2003–2012.

11. Titles of Figs 3b–3h are misleading, they should be responses not the variables themselves

*Response*: Sorry for the confusion. We have revised the titles in all figures accordingly.

12. Line 505: delete "also"

*Response*: Per your suggestions.

13. Line 512: "The aerosol influence on the monsoon, driven by the magnitude of aerosol–cloud interactions": How about aerosol–radiation interactions?

*Response*: Thanks for the suggestions. Unfortunately, without conducting additional experiments, it is difficult to quantitatively compare the impact of aerosol-cloud interactions and aerosol-radiation interactions. However, we can indirectly infer that aerosol-cloud interactions are likely more important from Figure R3 (Fig. S2 in the supplementary file). The SO$_2$ emission differences between CONT and CONTfA vary weakly between June and September (Figure R3b and R3f). Not surprisingly, the subsequent clear-sky downward shortwave radiation changes due to aerosol-radiation interactions show a similar pattern between June and September with minor changes through the season (Figure R3c and R3g). This suggests that the contrasting simulated aerosol-induced responses in precipitation, circulation, and temperature (Figure R3d and R3h) between June and September are likely primarily modulated by aerosol-cloud interactions as discussed in the main text. Furthermore, Dong et al. (2019) have conducted experiments to distinguish the effects of aerosol-radiation interactions and aerosol-interactions on the East Asian summer monsoon resulting from Asian aerosol changes using the MetUM HadGEM3 coupled model. They revealed that aerosol-cloud interactions play a predominant role in driving the overall circulation and precipitation responses. Given the limited space of the paper, we replace the word "driven" with

"modulated" in the revised text.

[Figure]

Figure R3. (a) The June climatological precipitation (mm day$^{-1}$) in CONT. June differences in (b) SO$_2$ emissions
(Tg yr$^{-1}$), (c) clear-sky downward shortwave radiation (W m$^{-2}$), and (d) near-surface temperature (K) between CONT
and CONTfA. (e–h) Same as (a–d) but for September.

**References**

Dai, L., Cheng, T. F., and Lu, M.: Anthropogenic warming disrupts intraseasonal monsoon stages and
brings dry-get-wetter climate in future East Asia, npj Climate and Atmospheric Science,
https://doi.org/10.1038/s41612-022-00235-9, 2022.

Dong, B., Wilcox, L. J., Highwood, E. J., and Sutton, R. T.: Impacts of recent decadal changes in Asian
aerosols on the East Asian summer monsoon: roles of aerosol–radiation and aerosol–cloud interactions,
Climate Dynamics, https://doi.org/10.1007/s00382-019-04698-0, 2019.

191 Wilcox, L. J., Dong, B., Sutton, R. T., and Highwood, E. J.: The 2014 hot, dry summer in northeast Asia,

192 Bulletin of the American Meteorological Society, 96, S105–S110, https://doi.org/10.1175/BAMS-D-15-

193 00123.1, 2015.

---

## Author Response (AR2)

1 Response to Review Comments

3 Dear Editor,

5 We thank the editor and the second reviewer very much for their further careful review

6 and valuable comments on our manuscript. We have tried our best to address all

7 concerns and revised the manuscript accordingly. Please note that the reviewer's

8 remarks are in black, our response is highlighted in blue, and extracts from the

9 manuscript are in red, with **new texts** that have been added/edited marked in bold. We

10 hope that you find the revision satisfactory. Thank you very much.

12 Kind regards,

13 Zhen LIU, on behalf of all co-authors

**Responses to Editor:**

After attending Laura's EGU talk, where she demonstrated that using only 3 ensemble members might not be sufficient, I am curious to hear your perspective on this matter in relation to the current paper.

*Response*: Thanks for the comments. We have analyzed the seasonal mean bias and response (JJAS) to Asian aerosols across different ensemble members (Figure R1). While sub-regional details vary from member to member, associated with increased Asian aerosols, all three ensemble members show a tilted dry/wet anomalous rainfall dipole over South Asia, with deficit rainfall over central India and wetting over the northwestern India and the northern Bay of Bengal. Additionally, all members display a wet/dry dipole over Eastern China (Figure R1d–R1f), with some inter-member differences in the orientation of the anomalies (e.g., from zonal bands in Ensemble 2 to northwest to southeast tilted anomalies in Ensemble 3). This general pattern of land precipitation anomalies bears a substantial resemblance to that in the ensemble mean (Fig. 3a–3c in the main text). Similarly, the seasonal mean bias in each ensemble member shows common features and close similarity to that of the ensemble mean. It is worth noting that while the three members are markedly similar over India, they show some differences over China. Ensemble 2 features an extensive zonal wet anomaly over southern China and a less pronounced, yet also zonally-oriented, dry anomaly to the north. The northern drying is more confined in Ensemble 1, which instead features anomalous wetting over central China. Finally, the drying is further confined to the eastern coastline in Ensemble 3, with further wetting inland. Comparing biases and aerosol-induced responses across the ensemble members it is possible to identify a link between their spatial patterns and inter-member similarities/differences. For example, the striking consistency of the bias pattern over India is associated to a similar response pattern in all ensemble members. Similarly, the more tilted anomalous bias in Ensemble 3 is closely similar to the corresponding response, as compared to the strongly zonal structure in Ensemble 2. Based on the mechanistic analysis of the Ensemble mean described in the manuscript, this suggests that the regional monsoon responses to Asian aerosol changes are strongly modulated by the regional precipitation biases: this link is consistent across different ensemble members and discernable despite the influence of internal variability. While this does not discard the possibility of internal variability to affect the monsoon response, the above comparison suggests biases to be crucial (and dominant) in determining the main features of the aerosol-driven changes. Note also that our analysis purposedly focuses on the externally-forced (ensemble mean) response, and not on quantifying its role against internal variability (e.g., as in detection and attribution studies), in light of the importance to better constrain the impact of human- made activities on the monsoon. In this regard, three (transient) ensemble members appear to be enough in our case, possibly because of the pronounced biases of the Met

Office model. The longer PDRMIP simulations, and related larger perturbations, still allow for the signal to emerge, despite some models display a smaller bias compared to the Met Office model. It will be interesting to extend this analysis to other models, for example in the context of the new RAMIP experiments.

[Figure]

Figure R1. JJAS precipitation (mm day$^{-1}$) bias (first row) and response (second row) to Asian anthropogenic aerosols (difference between CONT and CONTfA averaged during 2003–2012) in each ensemble member. Black dots mark grid-points for which the difference is significant at the 90% confidence level.

**Responses to Reviewer #2:**

1.  Fig. 1 evaluates the model performance in simulating precipitation and circulation, which shows the precip difference is between 6 to 10 mm/day vs. its climatology between 10 to 15 over Bay of Bengal, equivalent to ~50% of the climatology. This is a significant underestimation. I suggest the authors show the difference of precipitation in percent and perform a quantitative evaluation of model performance rather than a description of the difference.

*Response*: Thank you for the additional comments and suggestions. We have replaced the absolute precipitation differences with percentage differences (Figure R2c). Indeed, the model displays a considerable underestimation of precipitation amounts by 60% compared to observed climatological values over India and the Bay of Bengal. To the east, there are wet biases ranging from 20 to 60% of the climatology over southwestern China, and even up to 80% of climatological rainfall over the northwestern subtropical Pacific. We have included this figure and related description in the main text as follows:

[Figure]

Figure R2. June–September average precipitation (mm day$^{-1}$) and 850-hPa wind (m s$^{-1}$) for the observations (GPCP and CMAP average for precipitation, ERA5 for wind), the control simulation, and their differences (precipitation differences in percentage and wind differences in absolute values) during the period 1993 to 2012.

Lines 179–187: "Figure 1 compares the 1993-2012 June–September average precipitation and 850-hPa winds in the control simulation to observations (GPCP and CMAP average for precipitation, ERA5 for wind). The model reproduces the broad characteristics of the observed rainfall and circulation patterns (pattern correlation of 0.80 for precipitation, which is significant at the 99.9% confidence level). The difference panel indicates that the model **underestimates the rainfall amount by 60% over India and the Bay of Bengal** due to a weaker southwesterly monsoon flow. **To**

**the east, there are wet biases ranging from 20 to 60% of the observed climatology**

**over southwestern China, and even up to 80% over the northwestern subtropical**

**Pacific,** associated with enhanced cyclonic flow. Note that this bias pattern is common across CMIP6 models, although the magnitude of the anomalies varies from model to model (Wilcox et al., 2020), and is also consistent with that in the historical simulations of the CMIP6 Met Office model (Rajendran et al., 2022). A thorough discussion of the model bias and its linkage to regional and remote circulation can be found in Liu et al.

(2021)."

2.   And Figure R3 does not support that aerosol-cloud interactions play an more important role, because aerosol-cloud interaction in part means cloud responses to aerosols via aerosols working as cloud nuclei. Figure R3 does not shown cloud responses to aerosols. The contrasting meteorological patterns in June and September could be caused by many factors, and aerosol-cloud interactions are not the only candidate.

*Response*: Thank you for the comments. The differences between CONT and CONTfA

represent the total responses to changes in Asian anthropogenic aerosols, including both aerosol-radiation and aerosol-cloud interactions. As shown in Figure R3 (Figure R3 of previous revision), the aerosol-mediated changes in radiation show a similar pattern between June and September with minor changes through the season. This suggests that the contrasting aerosol-induced responses in precipitation and circulation (Fig. 5 in the main text) are likely modulated by aerosol-cloud interactions. The accompanied cloud responses to aerosols changes have been discussed in the Fig. 7 and Fig. 8 of the main text. We have toned down the role of aerosol-cloud interactions as the reviewer suggested in the first round of revision.

[Figure]

Figure R3. (a) The June climatological precipitation (mm day$^{-1}$) in CONT. June differences in (b) SO$_2$ emissions (Tg yr$^{-1}$), (c) clear-sky downward shortwave radiation (W m$^{-2}$), and (d) near-surface temperature (K) between CONT

and CONTfA. (e–h) Same as (a–d) but for September.